# The Germinal Origin of Salivary and Lacrimal Glands and the Contributions of Neural Crest Cell-Derived Epithelium to Tissue Regeneration

**DOI:** 10.3390/ijms241813692

**Published:** 2023-09-05

**Authors:** Hitomi Ono-Minagi, Tsutomu Nohno, Takashi Serizawa, Yu Usami, Takayoshi Sakai, Hideyuki Okano, Hideyo Ohuchi

**Affiliations:** 1Department of Cytology and Histology, Okayama University Graduate School of Medicine, Dentistry and Pharmaceutical Sciences, Okayama 700-8558, Japan; 2Research Fellow of Japan Society for the Promotion of Science, Tokyo 102-0083, Japan; 3Department of Cytology and Histology, Okayama University Medical School, Okayama 700-8558, Japan; 4Department of Physiology, Keio University School of Medicine, Shinjuku, Tokyo 160-8582, Japan; 5Department of Oral and Maxillofacial Pathology, Osaka University Graduate School of Dentistry, Osaka 565-0871, Japan; 6Department of Rehabilitation for Orofacial Disorders, Osaka University Graduate School of Dentistry, Osaka 565-0871, Japan; 7Department of Cytology and Histology, Faculty of Medicine, Dentistry and Pharmaceutical Sciences, Okayama University, Okayama 700-8558, Japan

**Keywords:** salivary and lacrimal glands, development, three germ layers, neural crest

## Abstract

The vertebrate body comprises four distinct cell populations: cells derived from (1) ectoderm, (2) mesoderm, (3) endoderm, and (4) neural crest cells, often referred to as the fourth germ layer. Neural crest cells arise when the neural plate edges fuse to form a neural tube, which eventually develops into the brain and spinal cord. To date, the embryonic origin of exocrine glands located in the head and neck remains under debate. In this study, transgenic TRiCK mice were used to investigate the germinal origin of the salivary and lacrimal glands. TRiCK mice express fluorescent proteins under the regulatory control of *Sox1*, *T/Brachyury*, and *Sox17* gene expressions. These genes are representative marker genes for neuroectoderm (Sox1), mesoderm (T), and endoderm (Sox17). Using this approach, the cellular lineages of the salivary and lacrimal glands were examined. We demonstrate that the salivary and lacrimal glands contain cells derived from all three germ layers. Notably, a subset of *Sox1*-driven fluorescent cells differentiated into epithelial cells, implying their neural crest origin. Also, these *Sox1*-driven fluorescent cells expressed high levels of stem cell markers. These cells were particularly pronounced in duct ligation and wound damage models, suggesting the involvement of neural crest-derived epithelial cells in regenerative processes following tissue injury. This study provides compelling evidence clarifying the germinal origin of exocrine glands and the contribution of neural crest-derived cells within the glandular epithelium to the regenerative response following tissue damage.

## 1. Introduction

The development of the digestive tract begins with a single tube as the endoderm folds to form the foregut and hindgut. The primitive intestinal tract consists mainly of an endoderm and a surrounding mesoderm, into which neural crest cells infiltrate to create the neural plexus. From this single tube, the digestive organs, from the pharynx to the upper anal canal, liver, and pancreas, are formed [1]. Mammals, including humans, also possess various smaller exocrine glands, including salivary glands for saliva production, a lacrimal gland (LG) for tear production, sweat glands for sweat production, and mammary glands for milk production [2]. The major salivary glands, including the parotid gland (PG), submandibular gland (SMG), and sublingual gland (SLG), originate from localized thickening, invagination, and outward extensions of the primitive oral cavity. A widely held view proposes that the coexistence of both salivary glands and LG within any given region implies that these exocrine glands are composed of a common progeny derived from the ectodermal germ layer, specifically surface ectoderm [3]. However, the progenitor cells of the salivary glands are also reported to be of endodermal origin, as they can differentiate into Cytokeratin17-expressing hepatopancreatic lineage cells [4]. On the other hand, the teeth are derived from the endoderm in certain vertebrates [5]. The demarcation between ectoderm and endoderm within the oral mucosa is considered significant for tooth and salivary gland positioning. It has long been believed that the SMG and SLG are of endodermal origin [6]. However, our understanding of the germinal origin of the oral epithelium has been limited owing to the challenges in distinguishing between ectoderm and endoderm during posterior mouth development [6]. One accepted endoderm marker is Sox17, but *LacZ* expression was not detected in the salivary glands of *Sox17*-2A-iCre/R26R mice [7]. In contrast, salivary gland defects are observed in ectodermal dysplasia syndrome, and several findings suggest an ectodermal origin of this disease [7,8]. Ocular tissue morphogenesis is a dynamic process involving intricate interactions between the epidermis and dermis. The presence of neural crest-derived cells in salivary gland tissue has been noted [9], but their significance has not been elucidated. 

The LG comprises multiple epithelial structures located in the external upper portion of the conjunctiva of the eyelid [10]. Elongation and branching of multiple epithelial buds from this region of the eyelid conjunctiva contribute to LG formation [11]. The LG is thought to be of surface ectodermal origin, but this has not been meticulously demonstrated [12]. In addition to the LG, some reports have examined the relationship between salivary glands and neural crest-derived cells [12,13,14]. Neural crest-derived cells are well known to possess a notable regenerative ability after injury in several organs, but to date, no reports discuss the contribution of neural crest-derived cells to the LG and the salivary gland injury models.

To investigate the germinal origin of salivary glands and the LG, the genetically engineered TRiCK mouse line was employed [13]. In this transgenic mouse, the neuroectoderm is visualized by a *Sox1*-driven fluorescence signal, the mesoderm by *T/Brachyury*, and the endoderm by *Sox17* [13]. Sox1 expression is specific to neuroectoderm and some populations of neural crest cells [14,15]. There are Sox1-positive cells in the neural crest border region of the ectoderm. Sox1 is known to be positive in not all but several populations of neural crest cells [16]. The T/Brachyury gene exerts a conserved function in establishing midline mesodermal structures in bilaterally symmetric animals, thereby determining the anterior-posterior axis [17]. Expression of this gene is crucial for mesoderm formation and differentiation into mesodermal lineages [18]. Sox17 is transiently expressed during extra-embryonic endodermal development and plays a central role in determining the fate of human primordial germ cells and digestive organs [19,20]. The primary objective of this study was to ascertain the origin of salivary and lacrimal glands using TRiCK mice.

## 2. Results

### 2.1. Analysis of Embryonic and Adult Salivary Glands of TRiCK Mice

Salivary glands develop within the primitive oral epithelium, invaginating the mesenchyme. Primordia of the three major salivary glands, SMG, SLG, and PG, were identified in the coronal section of fetal heads of embryonic day 12.0 (E12.0) mice (Figure 1A,B). Figure 1C–H depicts fluorescent protein expression in the salivary glands of E13.5 TRiCK mouse embryos. TRiCK mice have a three-color fluorescent protein-coding gene cassette, where Venus indicates *Sox1*-positive cell-derived tissue, mCherry indicates *T/Brachyury*-positive cell-derived tissue, and mCerulean indicates *Sox17*-positive cell-derived tissue. Three-dimensional (3D) images of the embryonic mice were captured as transparent whole-body pictures using a light sheet microscope (Figure 1C, Appendix A). The three colors labeled Venus, mCherry, and mCerulean exhibited no overlapping fluorescence signals. Furthermore, the axial section in the Z-series images of the E13.0 embryo appears to reveal a large number of Venus-positive cells were present in the prospective SMG and SLG (Figure 1D). In contrast, mCherry was primarily expressed in the mesenchyme. Most of the cells were mCerulean-negative, but a very few positive cells were present (Figure 1E–H). The expression of each fluorescent protein (FP) was confirmed in adult mice using frozen sections of SMG from postnatal day 50 (P50) TRiCK mice. These three colors were also observed in adult SMG tissues (Figure 1I–M). Morphological analysis revealed more intense fluorescence around the ducts. The expression of all three FPs in both fetal and adult tissues suggests that the SMG comprises tissues originating from each of the three germ layers. Primordia of the SLG and PG, which are major salivary glands other than the SMG, exhibited thickening of the oral epithelium with a similar expression pattern of the FPs to that observed in the SMG (Appendix A). The area percentages of each fluorescence-positive tissue in adult salivary gland tissue show that Venus-positive cells are the largest area with the analysis of Image J (Figure 1N). Since Keratin 14 (Krt14) has long been considered a biochemical marker of the surface ectoderm that develops the epidermis, we further examined the distribution of *Krt14*-driven cells in the salivary gland epithelium using *Krt14-cre* mice [21,22,23]. tdTomato-labeled *Krt14*-driven cells were widely distributed in the SMG and SLG glandular epithelia (Figure 1O,P), while several epithelial cells did not express tdTomato (Figure 1Q,R).

#### 2.1.1. Analysis of Salivary Gland Tissue Derived from Sox1-Positive Cells 

Because cells derived from all three germ layers (ectoderm, endoderm, and mesoderm) were found in salivary gland tissue, the cell types of adult salivary gland tissue were further investigated. To identify the properties of cells, especially focused on *Sox1*-dependent Venus-positive cells, co-staining was performed with a neuronal marker beta3-tubulin (TUBB3) (Figure 2A–D). The majority of Venus-positive cells were presumed to originate from nerves branching off from the SMG. In the magnified image, these cells expressing TUBB3 localized to the surface of the epithelia. However, there were regions where TUBB3 and Venus did not co-localize (indicated by the arrowheads in Figure 2B–D), where only Venus was detected. These Venus-positive, TUBB3-negative cells looked like epithelial cells. Morphologically, these Venus-only-expressing cells exhibited features distinct from those of nerves. Venus-positive cells were randomly found in the epithelium of the acinus (Figure 2E,F). Co-staining with antibody against an acinus marker water channel aquaporin 5 (Aqp5) confirmed Venus-positive cells in the acinus, indicating the presence of Venus-positive cells in some acinar cells (Figure 2F). These Venus-positive cells were also found in the duct epithelium and scattered along the line of the duct (Figure 2G,H). Co-staining with antibody against a duct marker Keratin 7 (Krt7) confirmed that Venus-positive cells in some duct cells (Figure 2H). Thus, *Sox1*-positive cell-derived epithelial cells were distributed throughout the salivary glands. Similarly, in the SLG and PG, we found cells with three germ layer origins and Venus-positive epithelial cells that did not co-express TUBB3 (Appendix A).

#### 2.1.2. Analysis of Salivary Gland Tissue Derived from T/Brachyury-Positive Cells 

The tissues other than epithelium include muscle, nerve, and connective tissues. Therefore, myoepithelial, neural, and vascular components were examined to evaluate the localization of *T/Brachyury*-positive cell-derived cells in adult glands. Co-staining with antibody against a myoepithelial marker, smooth muscle actin (SMA) was performed. SMA was localized in the vicinity of mCherry-positive cells, but they were not co-localized (Figure 2I). TUBB3 and mCherry were not co-localized as well (Figure 2J). Co-staining with an antibody against a blood vascular marker, CD31, was performed. Not all, but some mCherry-positive cells were positive for CD31. However, higher magnification of the image showed a membrane antigen CD31 was not co-localized with cytoplasmic mCherry fluorescence (Figure 2K). Thus, the mesoderm-derived tissue differentiated into the vascular tissue of the salivary gland.

#### 2.1.3. Analysis of Salivary Gland Tissue Derived from Sox17-Positive Cells 

Immunofluorescence was performed to know the property of *Sox17*-expressing cell-derived tissues. Co-staining with antibody against E-cadherin (E-cad), an epithelial adhesion marker, confirmed that *Sox17*-expressing cell-derived tissues, mCerulean-positive cells did not form the epithelium (Figure 2L). Furthermore, co-staining with antibody against TUBB3, CD31, and SMA showed mCerulean-positive cells did not express these three proteins. Thus, mCerulean-positive cells did not form epithelial cells, nerves, blood vessels, or myoepithelium (Figure 2L–O). These results showed that there were a few mCerulean-positive cells in the mesenchyme and those *Sox17*-expressing cell-derived tissues differentiated into mesenchymal tissues other than nerve, myoepithelium, or blood vessels in the adult SMG. There is a possibility that the *Sox17*-expressing cell-derived tissues may be related to incomplete recombination by *Sox17*-driven phiC31o, as reported previously [13]. 

### 2.2. Analysis of LG Tissue Showed Involvement of All Three Germ Layers

The LG is an exocrine gland in the head and neck region that originates from the prospective eyelid conjunctiva, similar to the developing salivary glands (Figure 3A,B). After rendering the whole LG tissue transparent, all three fluorescent protein markers were identified, implying the presence of LG cells originating from each of the three germ layers (Figure 3C–G). Notably, *Sox1*-dependent Venus-positive cells were found in both neural and epithelial tissues judging from their morphology. These Venus-positive cells were present in and near the acinar epithelium (Figure 3H) and duct epithelium (Figure 3I,J). Three-dimensional reconstruction of the confocal images revealed scattered epithelial cells forming the duct (Figure 3(I-1), Appendix A). When Venus-positive cells were observed in the glandular portion, they had cytoplasm with morphologies distinctly different from nerves (Figure 3(H-2,I-2)). Venus-positive cells were also irregularly distributed among the glandular cells. High-magnification views of the duct area revealed that the cells were distributed in a ring-like pattern (Figure 3(I-1,J-1,J-2)). Optical sectioning of the 3D images revealed that the duct was composed of two layers of cells, with a long nucleus parallel to the duct on the inner side and a nucleus perpendicular to the duct on the outer side, with the Venus-positive cells in the outer layer (Figure 3I). We also found differences in the frequency of Venus-positive epithelial cells between individual mice (Appendix A). However, there was no difference in the frequency of appearance between the left and right sides of individual mice, as verified by the X^2^ test (*p* = 0.60).

### 2.3. Involvement of Neural Crest-Derived Epithelial Cells in Tissue Repair

Sox1 is a known marker of neuroectoderm [24], and some neural crest cells transiently express Sox1 [25]. We thus hypothesized that the Venus-positive epithelial cells in the SMG and LG are neural crest-derived cells. We first examined whether Venus+ cells were found in tissues, such as the adrenal medulla, which contains cells of neural crest origin. We confirmed that the TRiCK adrenal gland contained Venus-positive cells (Appendix A), suggesting that the Venus+ epithelial cells in the SMG and LG are likely of neural crest origin. On the other hand, these data indicate that the number of Sox1-positive neural crest-derived cells was quite a small population. Neural crest-derived cells have been shown to contribute to cellular tissue regeneration via their stem cell properties [26]. We thus hypothesized that these *Sox1*-dependent Venus+ epithelial cells might play specific roles in adult glands, such as tissue regeneration and injury healing.

To investigate this hypothesis, we dissociated these exocrine gland cells into single cells and collected the epithelial cells using FACS (Appendix A). The differences in RNA expression between the two groups of cell populations, one with strong Venus expression and the other with weak Venus expression, were analyzed using quantitative PCR (qPCR). First, the expression level of a neural crest marker *Sox10* was compared between the two groups: *Sox10* expression in the cells with strong Venus increased more than the weak Venus cells (Figure 4A). We examined the expression of stem cell markers in the epithelial cells during salivary gland regeneration [27]. Expression levels of *Keratin5 (Krt5)*, *Axin2*, and *Bhlha15* (equal to *Mist1*) were significantly higher in the strong Venus+ cell population than in the low Venus+ cell population. Although expression levels of *Sox2* and *Kit* were not significantly different, high Venus+ cells tended to be higher than low Venus+ cells, as with other genes. A similar tendency in the expression of these three markers, which were significantly upregulated in strong Venus+ cells, was found in both SMG and LG tissues (Figure 4B and Appendix A). 

The behavior of these neural crest-derived cells in the LG was further examined in gland injury models (Appendix A). Two models were employed: a duct ligation model in which the salivary gland duct was ligated with silk threads and a mouse wound model in which the central region of the gland was damaged with a biopsy punch. Prior to using TRiCK mice, preliminary experiments were conducted on non-transgenic salivary glands (Appendix A). Sampling at various time points after duct ligation, release, and subsequent days revealed gradual restoration of damaged tissue to its normal state (Appendix A). Periodic Acid Schiff (PAS) staining showed progressive improvement in the production of mucus mucins, indicating the recovery of salivary gland secretory function after injury (Appendix A). Immunohistochemical analysis demonstrated the highest expression of *Krt5* seven days after ligation, prior to release, whereas expression of *Sox2* (another stem cell marker) was observed in the SLG but not in the SMG (Appendix A). These results are consistent with previous reports [28]. qPCR analysis validated the upregulation of *Krt5* expression seven days after release, consistent with our histological findings (Appendix A). 

Because *Krt5* upregulation was confirmed seven days after ligation in wild-type (WT) mice, we performed similar surgeries on LGs from TRiCK mice and examined them at the time points determined. Genotyping of mouse tails and optic nerves confirmed that individual mice had *Sox1*-dependent Venus-positive cells. Then, the duct ligation model was conducted in six animals, with one side ligated and the other side serving as a control. Ligated LGs exhibited atrophic gland clusters compared with the control side. When duct ligation was performed in a previous study to check molecular biological protein expression over time, a progenitor marker Krt5 expression was most increased on day 7 after ligation [29]. Therefore, the difference of LG was quantified on day 7 after ligation in this study. Analysis of damaged tissues in the ligated LG indicated an increase in Venus-positive cells compared to the controls (Figure 4C–I). As shown in Appendix A, Krt5, a myoepithelial marker [30], was present surrounding the Venus-positive acinar cells but also distributed around Venus-negative cells in both ligated and control LG (Figure 4E,H). Venus-positive cells were significantly increased in ligated LG compared to controls (Figure 4I). In the wound (punched) model, the size of the injured area was not clearly defined, although it appeared smaller after 8 days. Fluorescence microscopy revealed areas of Venus-positive cell accumulation within the injured region (Figure 4J–O). Experiments of regeneration from punched LG were performed on six mice. Although the two animals exhibited a clear increase in Venus-positive cells in the tissue, the overall results were comparable to those of the control SMG (Appendix A). The number of Venus-positive cell spots per field of view showed no significant difference between the punched LGs and control sides (Appendix A). In both SMG injury models, the samples displayed an elevated number of Venus-positive cell spots in the central region of the tissue (Appendix A). These findings suggest that *Sox1*-dependent Venus-positive cells, probably neural crest-derived cells, in the adult SMG contribute to the healing of damaged gland tissue.

## 3. Discussion

This study revealed that salivary and lacrimal glands trace their developmental origins to each of the three germ layers. In these tissues, embryonic origin is not uniform, and each of the three embryonic cell types contributes to specific functions. These exocrine glands most likely also contain neural crest-derived epithelial cells, which may contribute to tissue damage repair.

Teeth develop from the primitive oral epithelium, similar to the salivary glands. Many reports have investigated the developmental origin of teeth and found that they are derived from mixed ectoderm and endoderm [31,32]. The present study shows that the mesoderm-derived cells found in the oral mucosa are blood vessels [32]. Differences in the distribution of *Sox1/T/Sox17*-dependent fluorescence cells among individual TRiCK mice were also observed. In particular, some of the *Sox1*-dependent Venus-positive cells were found in the epithelium, whereas others were expressed only in the nerves. The expression pattern of Venus also differed between organs, with left–right differences observed. These results are consistent with those of a study using a *Sox1*-transgenic mouse line, in which expression of *Sox1* was observed in the head and neck epithelium, with individual differences [33]. Increasingly, genes that are differentially expressed across the left–right body axis appear to be influenced by differences in the microenvironment [34]. In the present study, we found scattered epithelial cells derived from the *Sox1*-dependent Venus-positive population, which was mostly neuroectoderm, in adult salivary and lacrimal glands. A surface ectoderm marker Krt14 is expressed in the developing eyelid conjunctiva, LG, oral mucosa, and salivary gland epithelia in mice [35,36]. In this study, the distribution of Krt14 was examined using *Krt14-cre* mice. While the majority of cells were tdTomato-positive, derived from the Krt14-expressing population, some cells were negative. These tdTomato-negative cells seemed to be of neural crest origin, but further identification should be performed with neural crest-derived cell markers in exocrine glands [37,38].

Early developmental neural crest cells are believed to possess stem cell-like properties [26]. Recently, the significance of neural crest-derived cells has been recognized not only in embryonic development but also in various aspects of aging, metabolism, and maintenance of the cardiovascular and motor systems. These findings highlight the involvement of these cells in various pathological conditions [39,40]. Neural crest cells gradually differentiate as they migrate throughout the embryo. During migration, neural crest cells send signals to and receive signals back from nearby cells and tissues [41]. The presence of a large number of neural crest-derived cells among the cell population assembled in response to tissue injury from duct ligation also indicates a high degree of migration. Further exploration is warranted to elucidate the characteristics of the dynamic distribution patterns of these probable neural crest-derived cell partners by investigating the distinctions between Sox1-positive/negative cells. Individual differences may paradoxically mean that Sox1-positive neural crest-derived cells are not directly involved in salivary gland histomorphogenesis. This suggests that cell differentiation is fluid and flexible. Neural crest-derived cells are expected to be a new cell source for regenerative medicine because of their multilineage potential even after growth [42]. As a few reports describe neural crest-derived epithelium in salivary gland tissue [43], it is expected to be utilized for epithelial tissue regeneration. Takahashi et al. showed double-coded neural crest-derived markers in transgenic mouse lines [44], supporting findings that neural crest-derived cells are, in fact, present beyond the individual differences shown in this study and that they may be a useful cell source for regenerative medicine.

In addition to the major salivary glands, we analyzed LGs in this study. Both the LG and the major salivary glands are exocrine glands of the head and neck, and in most mammals, they are primarily serous glands. Their branch pattern was classified as that of a compound tubuloacinar gland [45]. Clinically, these glands share many similarities, such as a similar pathology of malignancy, and both are targets of autoimmunity in Sjögren syndrome [46]. Among the exocrine glands of the head and neck, the salivary glands and LGs exhibit distinct development: the LG arises from the presumptive fornix superior conjunctivae, while the salivary glands originate from the primitive oral epithelium. Ocular tissue consists of surface ectoderm, neuroectoderm, neural crest, and mesoderm-derived cells [39,47]. The conjunctival epithelium and LGs are thought to originate from surface ectoderm. Although the periocular mesenchyme, thought to be derived from neural crest cells, comprises the connective tissue surrounding the glandular epithelial cells, this study provides the novel finding that neural crest-derived cells join the salivary and LG epithelia [48,49]. The reason for the higher frequency of neural crest-derived cells in LG than in the salivary gland requires further investigation. Although our data suggest that Venus-positive cells are involved in tissue regeneration, there was no significant increase in Venus-positive cells in SMG tissue after ligation, which might be attributed to the fact that Venus-positive tissue is less common in the SMG.

This study had certain limitations. First, the identity of *Sox17*-dependent mCerulean-positive cells (putative endodermal cells) was not thoroughly investigated. While cells in the endoderm give rise to digestive tract-associated glands and the exocrine pancreas, the involvement of the endoderm in the salivary and lacrimal glands remains unclear. During vertebrate development (e.g., zebrafish, chickens, and mice), endoderm and mesoderm cells arise from the same mesendodermal region. However, a complete understanding of endoderm cell migration and subsequent organogenesis is still lacking because not all cells derived from the endoderm exhibit the same behavior [50]. Thus, further detailed analyses are required to characterize *Sox17*-dependent fluorescent cells in TRiCK mouse salivary and lacrimal glands. Another limitation pertains to the individual variance observed among mice. Variations were noted in the frequency of Venus-positive cell appearance in the epithelium of the mice used in this study, as well as in the appearance of neural crest-derived cells in SMG injury experiments. One possible explanation for these differences is that the labeling of *Sox1*-dependent Venus-positive cells may not be complete due to insufficient Cre recombination [13]. Nevertheless, the mice used in this study exhibited fluorescent signals in the optic nerve, confirming the Venus labeling of retinal ganglion cells in a *Sox1*-dependent manner. Hence, inter-individual differences may exist in the distribution patterns and contribution rates of *Sox1*-positive neural crest cells. It remains uncertain whether differences in tissue regenerative capacity can be attributed to variations in the distribution of *Sox1*-dependent Venus-positive neural crest cells. To clarify this, it is necessary to investigate lineage-tracking lines that can complement the unlabeled cells. Detailed studies to identify *Sox1*-dependent Venus-positive cells using neural crest-derived cell markers will be a challenge in the future.

In conclusion, this study suggests that these exocrine glands originate from all three germ layers and that neural crest-derived epithelia are present in the epithelium and duct cells. Neural crest-derived tissues act as a source of cells for tissue damage repair and regeneration. These findings will advance our understanding of gland regeneration mechanisms and serve as a foundation for future regenerative medicine approaches aimed at the functional reconstruction of salivary and lacrimal glands. 

## 4. Materials and Methods

### 4.1. Mice

TRiCK mice were produced by crossing a reporter line (RBRC10966) that expresses fluorescent signals (Venus/mCherry/mCerulean) under the control of site-specific recombinases, with a germ layer labeling line (RBRC10967) bearing Cre under the control of the ectoderm marker *Sox1*, Dre under the control of the mesoderm marker *T* (*Brachyury*), and phiC31o under the control of the endoderm marker *Sox17*. Knock-in mice were purchased from RIKEN BioResource Research Center (Ibaraki, Japan). Pregnant *ROSA^MultiFPsΔPuro/MultiFPsΔPuro^* reporter females were crossed with the triple lineage-labeling males (*Sox1^2A-CreΔPuro/+^*, *Brachyury^2A-DreΔPuro/+^*, and *Sox17^2A-phiC31oΔPuro/+^*) and dissected at different days post-coitum. Venus expression was driven by *Sox1^2A-CreΔPuro^*, mCherry expression was driven by *T/Brachyury^2A-DreΔPuro^*, and mCerulean expression was driven by *Sox17^2A-phiC31oΔPuro^*. 

*Krt14-cre* mouse was obtained from Tg (*KRT14*-cre)1Amc/J (Jackson stock number 004782) [23] and B6.Cg-Gt(ROSA)26Sor^tm14(CAG-tdTomato)Hze^/J (Jackson stock number 007914) line were mated.

The ligation model, excluding TRiCK mice, was performed with female C57BL/6 mice. Histological analyses of the development of the salivary glands from the oral epithelium of the fetus used E12.0 C57BL/6-J mice, and analyses of the development of the LG from the eyelid conjunctival epithelium used E13.5 C57BL/6-J mice.

### 4.2. Mouse Injury Model of Duct Ligation

All animal surgeries were performed under the administration of three anesthetic mixtures, with the utmost effort made to minimize pain. The three anesthetic mixtures consisted of 0.75 mg/kg demi Thor (Zenoaq, Koriyama, Japan), 4 mg/kg midazolam (Sandoz, Tokyo, Japan), and 5 mg/kg Bettlefar (Meiji Seika Pharma, Tokyo, Japan). The anesthetic mixture was administered subcutaneously or injected into the abdominal cavity at a dosage of 0.1 mL per 10 g of mouse weight. After completing the surgery, an injection of 0.1 mL per 10 g mouse weight of a medetomidine antagonist-preparation liquid containing 0.75 mg/kg anti-sedatives (Zenoaq) in saline was administered into the abdominal cavity. Unilateral excretory duct ligation was performed as follows: P30 mice were anesthetized by intraperitoneal injection of pentobarbital (0.2 mg/g mouse weight), and the main excretory duct on one side of the neck was dissected and separated from the surrounding connective tissue under a surgical stereoscope (Leica S9D, Wetzlar, Germany). The main excretory duct was ligated using surgical sutures. After one week, the ligation procedure was reversed in some mice. Specifically, surgical sutures were removed from the main excretory duct, and the incision was closed. After three days, seven days, or two weeks of recovery, the mice were anesthetized and euthanized, and their glands were harvested. TRiCK mice were evaluated seven days after ligation. Positive cell counts were performed in a single-blind manner.

### 4.3. Tissue Punch Wounded Mouse Model

Under triple anesthesia, male TRiCK mice aged 6 weeks were subjected to surgical exposure of the SMG and LG, followed by the creation of surgical wounds using a biopsy punch with a diameter of 2 mm (Natsume Seisakusho, Tokyo, Japan). The injured glands were treated with a double-layer cell sheet in the experimental group. The left salivary gland sustained damage and was designated as the experimental group, whereas the intact right salivary gland was designated as the control group. As per previous publications, glandular tissue was evaluated 8 days after injury infliction.

### 4.4. Tissue cleaning

Embryos and exocrine organs were fixed using 2% paraformaldehyde (PFA) (Nacalai tesque, Kyoto, Japan) in 5% D-sorbitol–PHEM buffer adjusted to pH 8.0 for 2 days at 4 °C. After fixation, samples were washed three times with PHEM buffer. For tissue clearing, samples were immersed in a permeabilization buffer composed of PHEM, 2% saponin, 5% D-sorbitol, and 0.05% NaN_3_ for 8 h. Samples were subsequently immersed in a mixture of permeabilization buffer and IMES solution (composed of Omnipaque350 (Daiichi-Sankyo, Tokyo, Japan), 2% saponin (Nacalai tesque), 20% monoethanolamine (Nacalai tesque), 0.1% TritonX-100 (Sigma-Aldrich, St. Louis, MO, USA), 5% D-sorbitol (FujifilmWako, Osaka, Japan), 2.5 mM EDTA (Dojindo, Kumamoto, Japan), and 0.05% NaN_3_ (Fujifilm Wako) at respective ratios of 2:1 and 1:1 for 24–48 h. Finally, samples were immersed in 100% IMES for 24–48 h. All clearing procedures were performed at room temperature. Twenty-four hours prior to imaging, samples were transferred to a mounting solution composed of Omnipaque350, 0.1% TritonX-100, 5% D-sorbitol, 2.5 mM EDTA, and 0.05% NaN_3_, adjusted to pH 8.0 and passed through a 0.22 μm pore size filter. These tissue-clearing protocols have been previously published [13].

The cleared embryos were observed as three-dimensional (3D) spatial data on UltraMicroscope Blaze™ (Miltenyi Biotec, Bergisch Gladbach, Germany) using a 4× lens and excited 445, 488, and 561 nm laser for mCerulean, Venus, and mCherry, respectively. Then, 3D reconstitution and digital sectioning were processed by Imaris (BITPLANE).

For tissue clearing in section, samples were immersed in permeabilization buffer (PHEM buffer, 2% saponin (Nacalai tesque), 5% D-sorbitol and 0.05% NaN_3_ (Nacalai tesque). After incubating with secondary antibody, slices were cleared in IMES solution for 20 min, followed by transfer to mounting solution for another 20 min. Finally, slices were mounted on glass slides in a mounting solution with a 0.3 mm silicon spacer.

### 4.5. Cell Sorting

SMGs and LG from male TRiCK mice were processed individually. Tissues were depleted of blood in animals under anesthesia and removed. Tissues were disaggregated by incubation with an enzyme solution for 1 h at 37 °C for enzymatic treatment. The enzyme solution consisted of 1.67 mg/mL Collagenase (Thermo Fisher Scientific, Waltham, MA, USA), 1.33 mg/mL Hyaluronidase (Nacalai tesque), and 1.67 mg/mL Dispase II (Thermo Fisher Scientific) in D-MEM/F12. Epithelial cells were isolated using a CD44 antibody [51,52]. These cells were washed three times with HBSS-H and labeled with an APC-conjugated CD44 antibody (dilution 1:500, BioLegend, San Diego, CA, USA) at 4 °C for 30 min. Large cells were filtered using a 40 μm Nylon mesh, and single cells were collected in fresh 0.5% BSA-DMEM medium. APC-positive cells were separated from the total cell population using a fluorescence-activated cell sorter (FACS Aria3, BD Biosciences, Bridgewater, NJ, USA). The number of positive cells was small, and it was difficult to ensure sample quality. APC-positive cells were divided into two groups based on their Venus fluorescence intensities, strong or weak, using GFP filters. SMG and LG cells were analyzed by quantitative reverse transcription polymerase chain reaction (RT-qPCR). 

### 4.6. RT-qPCR

For RT-qPCR, each pooled RNA sample was reverse transcribed to generate complementary DNA (cDNA). qPCR was performed using FastStart Essential DNA Green Master Mix (06402712001, Roche, Basel, Switzerland) in a LightCycler Nano System (Roche). The qPCR results for each sample were normalized to the expression of a reference gene encoding glyceraldehyde-3-phosphate dehydrogenase (*Gapdh*). The results are presented as normalized ratios, and the experiments were performed in triplicate. PCR analyses employed the set of synthetic primers listed in Appendix A.

### 4.7. Immunohistochemistry (IHC)

Frozen samples were cryosectioned to a thickness of 12 μm using a Tissue Tek Polar cryostat/microtome (Sakura Finetek, Tokyo, Japan). After thorough drying of the glass slides, antigen retrieval was performed by heating them in instant antigen retrieval H buffer (LSI Medience Corporation, Tokyo, Japan) at 100 °C for 5 min. The slides were washed with phosphate-buffered saline (PBS), followed by incubation with primary antibodies diluted in a solution containing 1× PBS and 8% protein concentrate from the M.O.M.™ Kit (Vector Laboratories, Newark, CA, USA) overnight at room temperature. The specific antibodies used were anti-TUBB3 (dilution 1:200; R and D Systems, Minneapolis, MN, USA), anti-SMA (dilution 1:200; Sigma-Aldrich), anti-AQP5 (dilution 1:200, Alomone, Laboratories, Jerusalem, Israel), anti-KRT7 (dilution 1:100, Abcam, Bristol, UK), anti-E-cad (dilution 1:200; Cell Signaling Technology, Beverly, MA, USA), and anti-CD31 (dilution 1:100; Cell Signaling Technology).

Formalin-fixed paraffin-embedded tissue sections with a thickness of 6 µm were subjected to deparaffinization, and antigen retrieval was conducted through autoclave heating at a temperature of 121 °C for a duration of 5 min in instant antigen retrieval H buffer (LSI Medience Corporation, Tokyo, Japan). Slides were washed with PBS-Tween 20 (0.1%). Samples were incubated with M.O.M. mouse Ig blocking reagent (Vector Laboratories), followed by incubation with primary antibodies in a diluent overnight at room temperature. The specific antibodies used were anti-SOX2 (1:200; Cell Signaling Technology), anti- KRT5 (1:200; Cell Signaling Technology), and anti-E-cad (1:100; Cell Signaling Technology). After washing with PBS, the tissues were incubated with Cy2-labelled donkey anti-goat, Cy3-labelled donkey anti-mouse, and Cy5-labelled donkey anti-rabbit or anti-rat IgG (1:100; Life Technologies, Carlsbad, CA, USA) for 3 h at room temperature in a diluent composed of 5% donkey serum containing 8% protein concentrate. To quantify the surface area occupied by Venus-positive cells, six sections were taken from independent control and ligated glands and analyzed using an LSM780 microscope (Zeiss, Oberkochen, Germany) and ImageJ (https://imagej.nih.gov/ij/index.html accessed on 29 August 2023).

### 4.8. Hematoxylin-Eosin (HE) Staining and PAS Staining

Samples were immersed in 4% PFA solution in PBS. Paraffin sections were prepared from the collected embryos and tissues using a standard method, followed by HE staining. To evaluate the changes in mucin content in the tissue over time due to ligation, a PAS staining kit (ScyTek Laboratories, Logan, UT, USA) was employed.

### 4.9. Statistics

Statistical analyses were performed, and data were presented as mean values and standard deviations. Paired *t*-tests were used to compare mean values between the two groups. To analyze comparisons over time in the ligation model, the Bonferroni correction was applied for multiple comparisons.

## Figures and Tables

**Figure 1 ijms-24-13692-f001:**
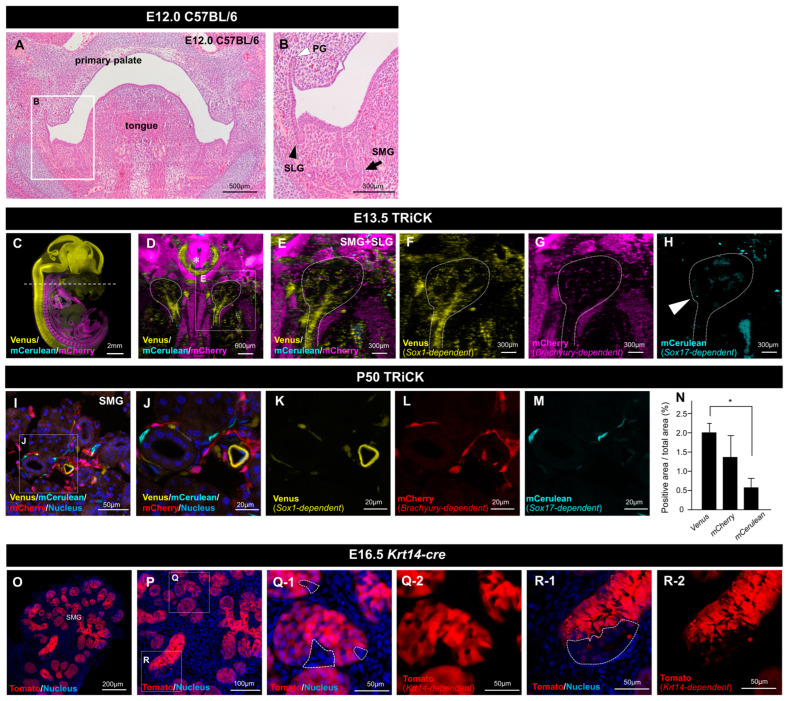
Fluorescently labeled salivary glands of TRiCK mice. (**A**,**B**) Histology (H&E staining) of a frontal section shows that major salivary gland formation has initiated in embryonic day 12.0 (E12.0) C57BL/6 mouse. The submandibular gland (SMG), sublingual gland (SLG), and parotid gland (PG) all begin morphogenesis within the primitive oral epithelium, invaginating into the mesenchyme. The boxed area in (**A**) is enlarged in (**B**). The black arrow shows SMG. The black arrowhead shows SLG. The white arrowhead shows PG. Scale bars: 500 μm (**A**), 300 μm (**B**). (**C**) The sagittal section image of the TRiCK embryo at E13.5, digitally reconstituted from light sheet microscopic observation, showed the simultaneous multi-colored labels in cells derived from all three germ layers. *Sox1^2A-CreΔPuro^* drove Venus (yellow); T/*Brachyury^2A-DreΔPuro^* drove mCherry (magenta); and *Sox17^2A-phiC31oΔPuro^* drove mCerulean (cyan). Scale bar: 2 mm. (**D**) Low-power image of an axial section at the height of the SMG and SLG (shown in dotted line in (**C**)). The dotted line shows the shape of the right and left SMG and SLG. Asterisk indicates trachea. Scale bar: 600 μm. (**E**–**H**) Images of the left SMG providing higher magnification views of (**D**). Scale bar: 300 μm. (**I**–**M**) Frozen sections of P50 TRiCK mouse SMG are shown under a confocal microscope with DAPI (**I,J**). Scale bars: 50 μm (**I**), 20 μm (**J**–**M**). (**N**) The percentage of each positive area in adult salivary glands is shown. Error bars indicate standard deviations. *n* = 5, * *p* < 0.05. (**O**–**R**) Frozen sections of E16.5 *K14 Cre* mouse SMG are shown. The boxed areas in (**P**) are enlarged in (**Q**,**R**). Red fluorescence (tdTomato)-negative epithelial cells were found at the height of the salivary gland (shown in dotted lines in (**Q-1**: with DAPI, **Q-2**: without DAPI) and (**R-1**: with DAPI, **R-2**: without DAPI)). Scale bars: 200 μm (**O**), 100 μm (**P**), 50 μm (**Q**,**R**).

**Figure 2 ijms-24-13692-f002:**
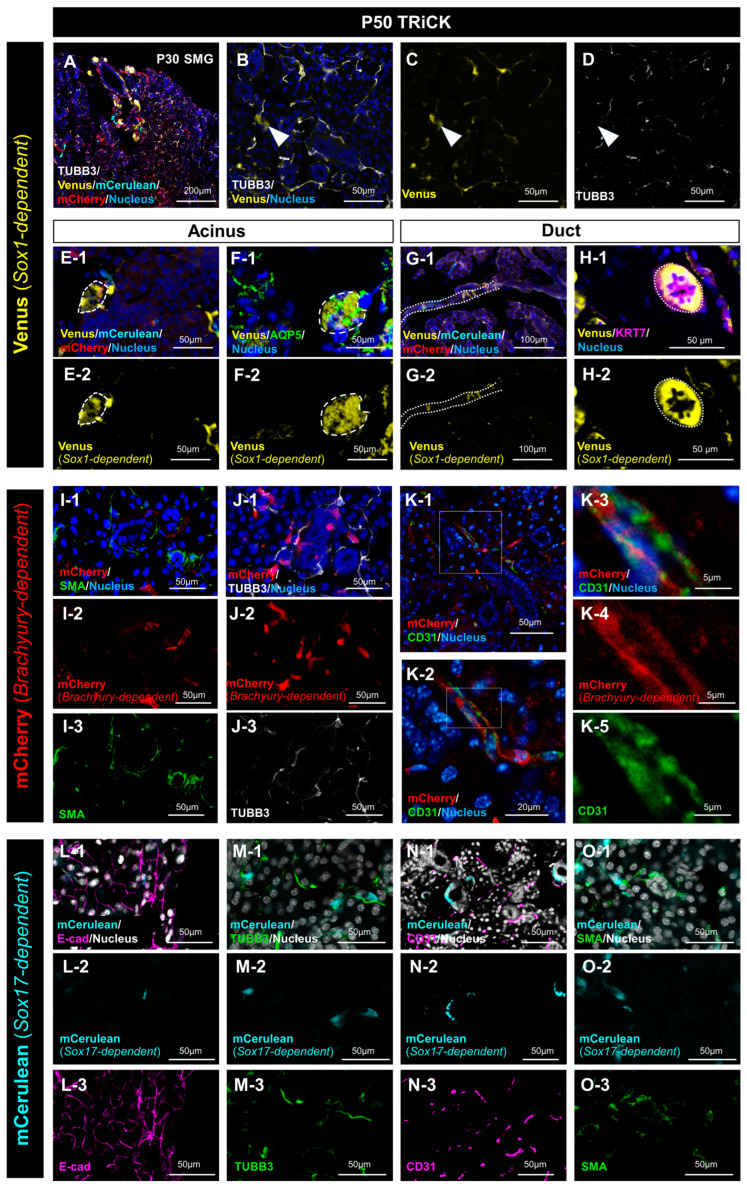
Profiles of the three germ-layer-derived cell populations in the adult SMG. (**A**–**D**) TUBB3 protein expression in the P50 TRiCK SMG was revealed by anti-TUBB3 staining and visualization under a confocal microscope. Merged view (**A**) shows five-color histology of DAPI staining and immunohistochemistry for TUBB3 in addition to the three-color fluorescence of TRiCK mice. The white arrowhead in (**B**-**D**) shows Venus-positive, TUBB3-negative cells. Scale bars: 200 μm (**A**), 50 μm (**B**–**D**). (**E**,**F**) Venus-positive cells in the acinus are shown. The area circled by the dotted line shows fluorescence signals in the cytoplasm of the acinar cells. (**F**) Aquaporin5 (AQP5) protein expression in P50 TRiCK SMG is shown by anti-AQP5 staining, and the images were taken by a confocal microscope. Upper panels (**E-1,F-1**) show merged images. Lower panels (**E-2,F-2**) show Venus-only images. Scale bars: 50 μm; (**G**,**H**) Venus-positive cells in the duct (dotted line) are shown. (**H**) Keratin 7 (KRT7) protein expression is shown by anti-KRT7 staining. Upper panels (**G-1**,**H-1**) show merged images. Lower panels (**G-2,H-2**) show Venus-only images. Scale bars: 100 μm (**G-1**,**G-2**), 50 μm (**H-1**,**H-2**); (**I**) alpha-smooth muscle actin (SMA) protein expression in P50 TRiCK SMG is shown by anti-SMA staining. Scale bars: 50 μm; (**J**) TUBB3 protein expression in P50 TRiCK SMG is shown by anti-TUBB3 staining. Scale bars: 50 μm. In (**I**,**J)**, upper panels (**I-1**,**J-1**) show merged images with nuclei in blue, while middle (**I-2,J-2**) and lower panels (**I-3**,**J-3**) show mCherry-only and SMA/TUBB3 localization-only images, respectively. (**K**) CD31 protein expression in P50 TRiCK SMG is shown by anti-CD31 staining. The boxed area in (**K-1**) is enlarged in (**K-2**). The boxed area in (**K-2**) is enlarged in (**K-3**–**K-5**). Scale bars: 50 μm (**K-1**), 20 μm (**K-2**), 5 μm (**K-3**–**K-5**). (**L**–**O**) Upper panels show merged images, while middle and lower panels show mCerulean-only and localization of E-cad/TUBB3/CD31/SMA only images, respectively. E-cadherin (E-cad) expression in the P50 TRiCK SMG is shown by anti-E-cad staining and observation under a confocal microscope. Scale bars: 50 μm (**L**). (**M**) TUBB3 protein expression in P50 TRiCK SMG is shown by anti-TUBB3 staining. Scale bars: 50 μm (**M**). (**N**) CD31 expression in P50 TRiCK SMG is viewed under a confocal microscope. Scale bars: 50 μm. (**O**) SMA protein expression in P50 TRiCK SMG is shown by anti-SMA staining. Scale bars: 50 μm (**O**). In all panels, frozen sections were used for analyses.

**Figure 3 ijms-24-13692-f003:**
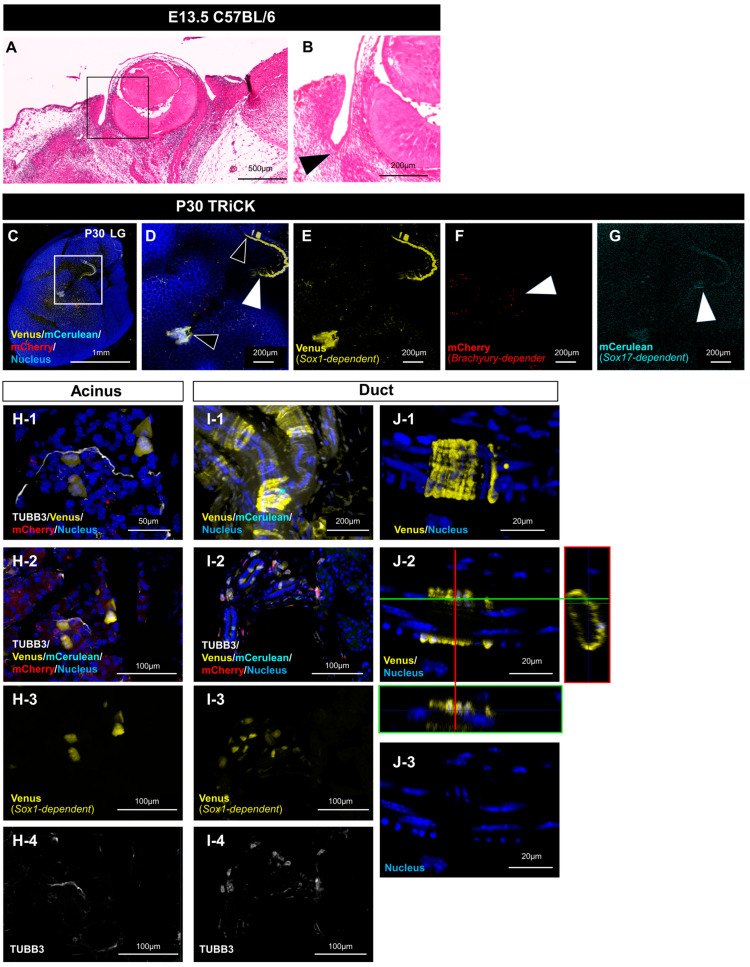
Profiles of the three germ-layer-derived cell types in the adult lacrimal gland (LG). (**A**,**B**) Developing eyelid conjunctiva in the frontal section. Histology shows that the salivary gland s emerging at E13.5. The boxed area in (**A**) is enlarged in (**B**). Scale bars: 500 μm (**A**), 200 μm (**B**). (**C**) LG from cleared TRiCK mouse tissue is shown with nuclei in blue (DAPI). Boxed area in (**C**) is enlarged in (**D**–**G**). The Venus-positive line (white arrowhead) shows a nerve. Black arrowheads show Venus-positive cells of the acinus and duct epithelia (**D**). White arrowhead shows mCherry-positive cells (**F**). White arrowhead shows mCerulean-positive cells (**G**). Scale bars: 1 mm (**C**), 200 μm (**D**–**G**). (**H**) The frozen section shows the distribution of Venus-positive cells in the acinus of the LG. Venus+ cells are TUBB3-negative. (**H-1**) Z-stack image constructed from acinus sections taken with a confocal microscope. Fluorescence image of a cleared LG is shown. (**H-2**–**H-4**) Zoom-out image of (**H-1**). Scale bars: 50 μm (**H-1**), 100 μm (**H-2**–**H-4**). (**I-1**) Z-stack image of the duct area taken with a confocal microscope, showing Venus-positive cells were distributed in a ring-like pattern. Scale bars: 200 μm. Its 3D image is shown in Appendix A. (**I-2**–**I-4**) Frozen section shows Venus+ cells in the duct area of LG are TUBB3-negative. Scale bars: 100 μm. (**J-1**) Z-stack image of the duct area in higher magnification. Scale bars: 20 μm. (**J-2**,**J-3**) Z-stack images in (**J-1**) were processed to a single plane using imaging software. The image on the *X*-axis, indicated by the green line in (**J-2**), is shown below, and the image on the *Y*-axis, indicated by the red line, is shown on the right. Scale bars: 20 μm.

**Figure 4 ijms-24-13692-f004:**
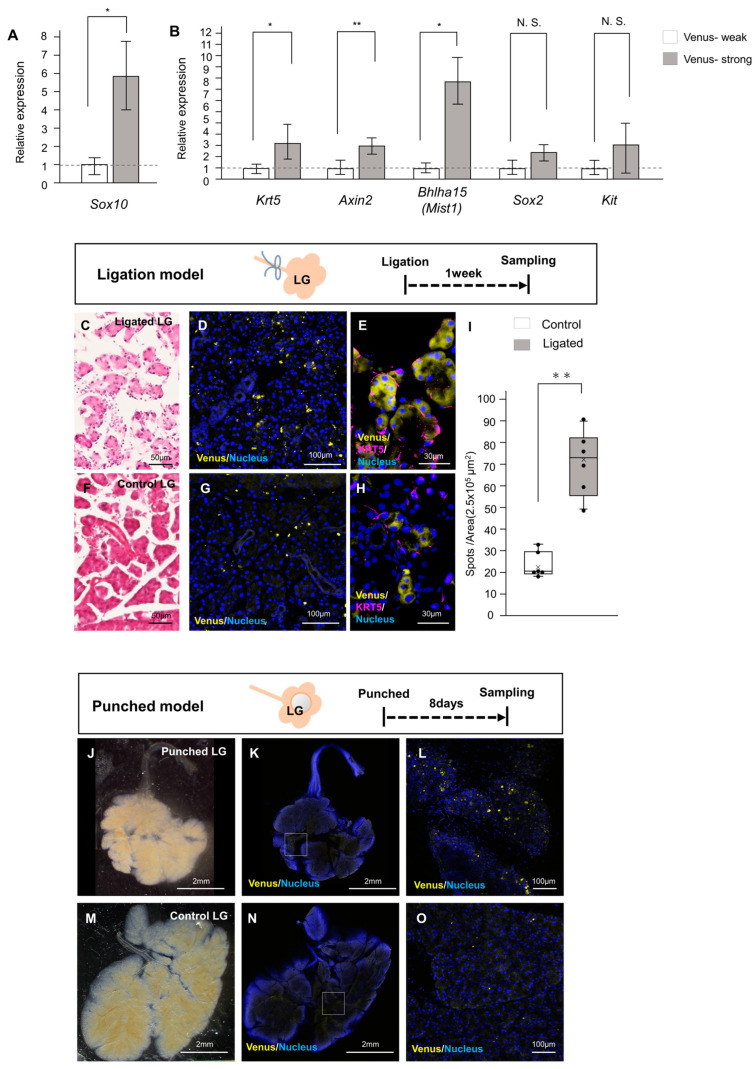
Contribution of *Sox1*-dependent Venus+ epithelial cells to LG tissue regeneration. (**A**) Comparison of a neural crest marker *Sox10* expression between Venus-strong and Venus-weak LG cells by qPCR. Error bars indicate standard deviations. *n* = 6, * *p* < 0.05 (**B**) Comparison of *Krt5*, *Axin2*, and *Bhlha15* (*Mist1*), *Sox2*, and *Kit* expression between Venus-high and Venus-low LG cells by qPCR. No significant (N. S.) differences were found between the two groups in *Sox2* and *Kit*. Error bars indicate standard deviations. *n* = 6, ** *p* < 0.01, * *p* < 0.05. (**C**–**H**) Histology and fluorescence images of TRiCK mouse LG at 7 days after ligation (**C-E**) and non-ligated control LG (**F**–**H**). (**E**,**H**) Krt5 protein expression in LG is shown by anti-Krt5 staining, and the images were taken using a confocal microscope. Scale bars: 50 µm (**C**,**F**), 100 µm (**D**,**G**), 30 µm (**E**,**H**). (**I**) The number of Venus-positive cell spots in one area from six individual mice showing that ligated LGs have a higher number of Venus-positive cells. Error bars indicate standard deviations. Dots indicate respective values. *n* = 6, ** *p* < 0.01 (**J**–**L**) Stereoscopic bright-field and fluorescent LG images at 8 days post-surgery. (**M**–**O**) Control LG. Boxed areas in (**K**,**N**) are enlarged in (**L**,**O**). Scale bars: 2 mm (**J**,**K**,**M**,**N**), 100 µm (**L**,**O**).

## Data Availability

The data that support the findings of this study are openly available.

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
