# Peer review of "The Germinal Origin of Salivary and Lacrimal Glands and the Contributions of Neural Crest Cell-Derived Epithelium to Tissue Regeneration"

_ijms, 2023, doi:10.3390/ijms241813692_

Round 1
Reviewer 1 Report
This paper identifies the epigenetic origin of salivary and lacrimal glands.
The introduction gives a detailed explanation of the genomic content, an understanding of the previous studies, and a reference citation explaining the validity and purpose of the study well.
However, to be accepted in this journal, the following points must be revised.
Results - Histological studies have well-supported intuitive genomic proofs by showing markers of triple lobes through fluorescence microscope imaging, but there are some points to be revised. Since most experiments are confirmed through histological staining, quantitative analysis of markers is overall insufficient.
1. The image alone seems to be insufficient to conclude the authors' claims. In Figure 1, it would be good to add molecular biological data on marker gene expression in embryonic and adult mice (qPCR, western blot etc).
2. In Figure 4, the authors claim that Sox1-dependent Venus-positive cells (probably neural crest-derived cells) contribute to the healing of damaged tissue in adult SMG. At this time, it would be good to add molecular biological expression data that can quantitatively confirm the increase and decrease of markers related to healing and regeneration for each day in the aspect of tissue regeneration and healing. It would be nice to add an intuitive picture of the healing aspects of each day. In addition, Figure 4A) explains that three stem cell markers are expressed high in Venus-high cells compared to low cells at RNA level. However, considering the expression of Axin2 and Bhlha15 except for Krt5, it seems that the error bar is large in Venus-high epithelial cells. In this regard, reducing errors through methods such as increasing the number of repetitions of experiments will increase the reliability of the results. Next to Figure 4 K/N, if you graph the fluorescence figures in the same way that graphed next to Figure 4 D/G, it helps to understand the results.
3. As most conduct mouse experiments, It is better to add a schematic figure, such as time point to experiment, will improve the understanding of the manuscript.
4. In Figure 5, the properties of several cells with the morphology of Venus-positive cells before confirming gene expression in certain cells and conducting subsequent experiments, but there seem to be too many figures about identifying cell types or profiles. Please check if there are any unnecessary duplicate figures for verification and reduce the amount that can be reduced.
5. It is a little too much to claim that the cells are stem cells and have the effect of wound healing just by checking the expression of Keratin5 (Krt5), Axin2, Bhlha15, and Sox2. The authors need to check additional stem cell markers and conduct other experiments that show the characteristics of stem cells, and then check more various things about wound healing/duct ligation.
Reviewer 2 Report
The study by Minagi-Ono et al describes the use of the TRiCK mouse model to determine the germ line source of the three major salivary glands and the lacrimal gland in mice. While the question is a very valid one and the findings would be an excellent addition to the field, the data presented to back up the conclusions is not very strong and a number of corrections must be made in order to make the manuscript of publishable quality, as outline below:
Major corrections
1. How efficient is the TRiCK mouse model? The authors should include this in the results and discussion. Furthermore, how do the authors explain cells that are not labelled by any colour in the submandibular gland (e.g. the large duct in Figure 1K and the acinar cells in the top part of the same image, which are marked by nuclei only). This is even more apparent in the SLG data (Supp Figure 1) where there are very few cells labelled with any of the three colours. This data demonstrates that the majority of cells in the three major salivary glands and the lacrimal glands are not derived from neuroectoderm, mesoderm or endoderm, or that the model is not ideal to answer the authors’ question.
2. The characterisation of Venus-expressing cells is not particularly convincing. Lineage markers are required to support the statements “Venus-positive cells were randomly found in the epithelium 138 of the acinus (Figure 2E, F). These Venus-positive cells were also found in the duct epithelium and scattered along the line of the duct (Figure 2G, H).” Please co-stain with AQP5, MIST1 or NKCC1 for acinar cells and KRT7, 8 or 19 for ductal cells.
3. The SMA staining in Figure 2 is not very convincing – it should look like the images displayed in PMIDs: 30305288 and 25238060.
4. The statement “Co-staining with antibody against E-cadherin (E-cad), an epithelial adhesion marker, confirmed Sox17 expressing cell-derived tissues formed the epithelium.” is misleading – the majority of epithelial cells in the Figure 2Q-T are Cerulean negative.
5. Are the authors convinced that the signal in Figure 3F and G is real? It doesn’t look very convincing that there is any contribution of T/Brachyury or Sox17 positive cells to the developing LG.
6. An epithelial cell marker should be used to support the statement “Notably, Sox1-dependent Venus-positive cells were found in both neural and epithelial tissues judging from their morphology.” (line 183)
7. Can the authors confirm the association of Venus+ cells with KRT5 expression, for example, at the protein level by co-staining for KRT5. There is a commercially available antibody (e.g. PMID: 28623666).
8. Does the statement “Although the two animals exhibited a clear increase in Venus-positive cells in the tissue..” on line 261 mean that only two animals were analysed in this experiment? If so these numbers must be increased as 2 mice is not sufficient for robust analysis.
9. The authors should quantify some/all of the data presented in this study (e.g. % of cells that are Venus+, Cerulean+, mCherry+ throughout timepoints and glands analysed). This will convince the reader of the strength of the conclusions of the study. This is particularly important given that the authors themselves state “Variations were noted in the frequency of Venus-positive cell appearance in the epithelium of the mice used in this study, as well as in the appearance of neural crest-derived cells in SMG injury experiments.”
Minor corrections
10. Please cite Farmer, et al. PMID: 28576768 when discussing LG development and the involvement of KRT5+ cells.
11. Please reference the result that shows the statement “Venus was also expressed in the mesenchyme of fetal salivary glands, whereas mCherry was primarily expressed in the mesenchyme”
12. Line 135: I suspect Figure 1A-D should be 2A-D
13. What is the basis for the statement “These Venus-positive, TUBB3 negative cells looked like epithelial cells.” on line 135.
14. What was the rationale for analysing myoepithelial cells/choosing SMA as a candidate for characterisation of T/Brachyury positive cells?
15. Please show which area in Figure 3C is magnified in Figure 3M.
16. Studies that have shown that Krt5, Axin2 and Bhlha15 are progenitor cells should be cited in line 231.
17. Please state that the statement “whereas expression of Sox2 (another stem cell marker) was observed in the SLG but not in the SMG (Figure S5E-G, S6E-G).” is in agreement with published data (PMIDs: 29335337 and 21982232).
18. Do the authors believe the following statement means that there is no contribution to repaired LG by Venus+ cells? “Although the two animals exhibited a clear increase in Venus-positive cells in the tissue, the overall results were comparable to those of the control SMG (Figure S3B- E).”
19. Please include sufficient data in the figure legends regarding number of mice analysed.
20. FACS data should be included to show the sorting strategy for qPCR analysis of different labelled cells.
The language is appropriate throughout.
Reviewer 3 Report
Manuscript ID: ijms-2566892
Manuscript Title: Germinal origin of salivary and lacrimal glands and contributions of neural crest cell-derived epithelium to tissue regeneration
Authors: Hitomi Minagi-Ono, Tsutomu Nohno, Takashi Serizawa, Takayoshi Sakai, Hideyuki Okano and Hideyo Ohuchi
In study, Minagi-Ono and colleagues characterize the development of the salivary glands and lacrimal gland in TRiCK mice in order to characterize the cell populations comprising these tissues and the germ layers which they arise from. The study is very interesting and bring about several, very-significant interesting insights like the identification of cells from all three germ layers in the salivary glands. The study itself is technically sound. However, the authors frame the study around neural crest cells and the neuroectoderm (Sox1-positive) cells in the tissues. While this would be an interesting mechanism, they just do not provide a credible link between these cell positions and neural crest cells. The evidence provided is circumstantial. Although the study is interesting, the authors should reframe the narrative of their manuscript to focus on the evidence at hand and not reach too far into inference. Below is a list of issues (in no order of significance) which the authors must address before this manuscript is ready to be published.
1. As mentioned above, the authors use several statements such as in the abstract “Sox1-driven fluorescent cells differentiated into epithelial cells, implying their neural crest origin”. It is unclear how this implies neural res cell origin. Many non-neural crest cells become epithelial.
Further in the Abstract, the authors state: “These cells were particularly pronounced in duct ligation and wound damage models, suggesting the involvement of neural crest-derived epithelial cells in regenerative processes following tissue injury.” again, using terms like “suggesting” without any evidence is incorrect. Many other tissues are involved in tissue injury healing, not just neural crest cells.
2. Lines 64 and 64: The authors write “Sox1 expression is specific to neuroectoderm and some populations of neural crest cells [14,15].” Two issues here. First, just because some populations of neural rest cells express Sox1 doesn’t mean that all Sox1 positive cells are of neural rest cell original. Second, reference 15 does not mention Sox1 as being of neural crest original, just that is is a neuronal marker.
3. Arrows to the salivary gland priomordia in Figure 1A and B would be very helpful.
4. The images in the manuscript itself are of very low resolution. This may be a byproduct of the software that compiles the pdf, but the authors should check on this.
5. Which salivary gland is depicted in Figures 1D-H?
6. For the results in Figure 2, the authors performed immunioflurescence with the neuronal marker beta3 tubulin (TUBB3) marker. This is a neuronal marker, the same as Sox1 so overall would not be surprising. If the goal of the study was to see if Sox1-positive cells are of neural crest origin, why not test for a neural crest marker like Sox10? If SOx10-positive cells overlap with Sox1 then the authors can more confidently claim that Sox1-positive cells are of neural crest origin.
7. Line 135 do the authors mean Figure 2 instead of Figure 1?
8. Lines 154-156. Is there a better image/region to use for Figure 2Q-T with more mCerulean?? there appears to only be one small signal of mCerulean and it doesn’t look like it overlaps with a specific cell, but rather is between cells, as can be seen in Figure 2Q. Not very representative or if it is, this is not enough to draw any conclusion. There definitely are a lot of cells that are not positive for endodermal mCerulean, but one cannot say the other way around.
9. Lines 156-157 The authors chose to co-stain with an antibody against CD31, a vascular endothelial marker. Vascular endothelial cells are mesodermal in origin, so not finding coexpression with mCerulean-positive cells would not be surprising right? considering that the mCerulean-positive cells are supposed to be endodermal? Why did the authors choose the CD31 marker??
10. Figures 3F and G are very dark making it hard to see any signal. The authors should either use arrows to point out the signal or make them brighter.
11. Lines 217-218. The authors write “Sox1 is a known marker of neuroectoderm [20], and a population of neural crest cells is positive for Sox1 [21].” As I understand Reference 21, it shows that neural tissues expressing Sox1 gradually loses Sox1 expression as it becomes neural crest cell. This does not mean that Neural crest cells express Sox1, but rather the opposite.
12. Along the same lines, in lines 218-220, the authors write “We thus hypothesized that the Venus-positive epithelial cells in the SMG and LG are neural crest-derived cells. We first examined whether Venus+ cells were found in tissues, such as the adrenal medulla, which contains cells of neural crest origin”.
Again, just because of the medulla has neural crest cells and Venus-positive cells doesn’t mean they are the same cells. The medulla has many other cells that are not Venus+ and maybe those are the neural crest cells?
Again, the argument for the Venus positive cells being neural crest cells is always collective and the authors do now who concrete evidence to prove this claim.
13. Lines 289-301. The authors discuss at length about differences on the left-right sides of the body which has very little to do with the experiments conducted here. It is unclear why so much space is spent on this topic.
Round 2
Reviewer 1 Report
Now it is acceptable
Reviewer 2 Report
The authors have addressed all my comments, thank you.
Reviewer 3 Report
The authors have addressed all of my concerns. I have no further comments.